# Microporous Formation Mechanism of Biaxial Stretching PA6/PP Membranes with High Porosity and Uniform Pore Size Distribution

**DOI:** 10.3390/polym14112291

**Published:** 2022-06-05

**Authors:** Wenxiang Fang, Guixue Liang, Jiang Li, Shaoyun Guo

**Affiliations:** The State Key Laboratory of Polymer Materials Engineering, Polymer Research Institute of Sichuan University, Chengdu 610065, Sichuan, China; 13008190145@163.com (W.F.); oolgxoo@163.com (G.L.); nic7702@scu.edu.cn (S.G.)

**Keywords:** microporous formation mechanism, isotactic polypropylene, PA6 spherical particles, PA6 fibers, biaxial stretching, microporous membranes

## Abstract

The low porosity and wide pore size distribution of biaxial stretching PP microporous membranes continue to be the primary impediments to their industrial application. To solve this problem, there is a critical and urgent need to study the micropore-forming mechanism of PP membranes. In this research, the interfacial micropore formation mechanism of PA6/PP membranes during biaxial stretching was investigated. PA6/PP membranes containing spherical PA6 and fibrillar PA6 were found to exhibit different interfacial micropore formation mechanisms. Numerous micropores were generated in the PA6/PP membranes, containing PA6 spherical particles via the interface separation between the PP matrix and PA6 spherical particles during longitudinal stretching. Subsequent transverse stretching further expanded the two-phase interface, promoting the breakdown and fibrosis of the PP matrix and forming a spider-web-like microporous structure centered on spherical PA6 particles. In PA6/PP membranes with PA6 fibers, fewer micropores were generated during longitudinal stretching, but the subsequent transverse stretching violently separated the PA6 fibers, resulting in a dense fiber network composed of PA6 fibers interwoven with PP fibers. Crucially, the PA6/PP biaxial stretching of microporous membranes presented an optimized pore structure, higher porosity, narrower pore size distribution, and better permeability than β-PP membranes. Furthermore, this study explored a new approach to the fabrication of high-performance PA6/PP microporous membranes, with good prospects for potential industrial application.

## 1. Introduction

Polyolefin microporous membranes (PP and PE) have been used for a wide range of applications in battery separators due to their low cost, excellent mechanical performance, chemical stability, and ease of manufacturing. Three industrial methods are used to produce battery separators, which are as follows: uniaxial stretching of polyolefin hard elastic films [1,2,3], biaxial stretching of β-PP [4,5], and wet biaxial stretching of UHMWPE [6,7]. The wet process is costly and unfavorable to the environment because of the need to extract mineral oil from the membrane. The dry uniaxial stretching technique involves annealing the cast film, which complicates continuous manufacturing. The biaxial stretching of β-PP does not require steps such as mineral oil extraction and annealing treatment, so it can be operated continuously with high efficiency, and creates no environmental pollution. However, the wide pore size distribution and low porosity of biaxial stretching β-PP membranes remain as challenges that limit their wide application [8,9,10]. To enhance the porosity and narrow the pore size distribution, research on the pore-forming mechanisms of β-PP membranes is necessary [11,12,13,14].

Over the past few decades, research has been conducted on the pore-forming mechanisms of β-PP during biaxial stretching [15,16,17]. The defect initiation mechanism, in which initial micropores are formed immediately by interfacial debonding of β crystals, has been acknowledged by the majority of researchers. They revealed that the polydispersity of β lamellae caused the β-PP membranes’ low porosity and large pore size distribution. Since the bundle-like β crystals are defective spherulites with an asymmetrical structure, only the β lamellae perpendicular to the stretching direction are separated by the external force, resulting in fewer early microcracks. This ultimately results in the development of coarse PP fibers, poor porosity, and a wide pore size distribution in β-PP membranes [5,18]. As a result, some researchers have attempted to overcome the disadvantages of β-PP membranes’ low porosity and wide pore size dispersion by manipulating the morphology of the β crystals [18,19,20,21,22,23]. Yang et al. [18] found that “flower-like” β-crystals and β-spherulites, rather than “bundle-like” β-crystals and β-hedrites, exhibited homogenous deformation and narrow pore size dispersion of membranes. Ding et al. [5] discovered that radial-growth β-spherulite produced abundant micropores with a narrow pore size distribution. Wu et al. [19] and Liang et al. [24] obtained β-PP membranes via biaxial stretching of β-transcrystallinity (β-TC) films. They concluded that numerous nearly parallel-aligned β lamellae in β-TC were more prone to separation and developed into more pores with a narrower pore size distribution under external forces. Overall, the ultimate objective of these approaches mentioned above is to separate more β-lamellae interfacial defects during stretching.

Additionally, by introducing weak two-phase interfaces to replace the β-lamellae defects for separation in the precursor films, the porosity and pore size distribution of stretched membranes can also be controlled effectively [25,26,27,28,29,30,31]. For example, in immiscible PP/PS and PP/PET stretched membranes, the micropore-forming mechanism of phase interface separation has been investigated [25,27]. The pore size distribution of these microporous membranes was reported to be quite narrow. However, they also exhibited extremely poor porosity and permeability, which significantly limited their applicability on a large scale. Feng et al. [30] fabricated PA6/PP blends with PPgMA as the compatibilizer, and sequential biaxial stretching was applied. They found that only membranes with a PA6 content of more than 50% can generate through pores. However, the introduction of excess PA6 changed the intrinsic qualities of the PP. More crucially, the effects of PA6 morphologies on the micropore-forming mechanism and microporous structure of PA6/PP membranes were not studied in detail.

On the one hand, the addition of the second phase needs to be reduced to minimize the influence on the intrinsic qualities of the PP membrane. On the other hand, it is necessary to carry out a systematic and in-depth study of the micropore-forming mechanisms of polymer blend microporous membranes during biaxial stretching, to provide theoretical guidance for the preparation and application of high-performance microporous membranes. Our previous work investigated the micropore-forming mechanism of PTFE fibers/PP membranes and demonstrated that PTFE fibers can improve the porosity and narrow the pore size distribution of membranes [28]. In this work, the effect of PA6 morphologies (spherical and fibrillar PA6) on the micropore-forming mechanisms and the performance of PA6/PP biaxial stretching membranes were explored. SEM and porosity tests were mainly used to track the origination and enlargement of micropores in PA6/PP membranes. The results showed that PA6/PP microporous membranes with different two-phase interface morphologies exhibited completely different micropore-forming mechanisms during the stretching process. Compared to β-PP membranes, PA6/PP biaxial stretching membranes showed a better pore structure, more uniform pore size distribution, enhanced porosity, and better permeability.

## 2. Materials and Methods

### 2.1. Materials

Isotactic polypropylene, T36F, was purchased from Maoming Petrochemical Co., Ltd. (Maoming, China). PA6, Ultramid B27 E, was purchased from BASF SE (Ludwigshafen, Germany). β-NA (TMB-5) was purchased from Shanxi Provincial Institute of Chemical Industry (Taiyuan, China). The antioxidant 1010 was purchased from Virtulla Tianjin Technology Co., Ltd. Polypropylene grafted maleic anhydride (PPgMA, grade GPM200B) was purchased from Ningbo Nengzhiguang New Materials Technology Co., Ltd. (Ningbo, China). The melt index of PPgMA is 30–50 g/10 min (230 °C/2.16 kg). The grafting rate of PPgMA is about 0.6–0.8 wt.%.

### 2.2. Sample Preparation

First, PP, PPgMA, PA6, TMB-5, and antioxidant 1010 were dried in a vacuum at 80 °C for 24 h. Then β-PP, PA6/PP(S), and PA6/PP(F) were produced using a twin-screw extruder. The temperatures profile was 185 °C, 225 °C, 235 °C, and 230 °C from hopper to die. The compositions of three samples are displayed in Table 1, where S and F represent the spherical and fibrillar PA6 phase morphologies, respectively. The compatibilizer, PPgMA, can react with PA6 to form the graft copolymer PP-g-PA6 [12,32]. It reduced the PA6 phase domain size and provided adequate adhesion between PP and PA6 to prevent film breakage during stretching.

To obtain the fibrillar PA6, the extruded melt was stretched using a three-roll machine with a stretching ratio (i.e., the ratio of the cross-sectional area of the unstretched samples to that of the stretched samples [33]) of about 100. The hot-stretched PA6/PP(F) sample was frozen in a cold-water bath before reaching the take-up roller. For PA6/PP(S), the stretching ratio was 1 and the PA6 dispersed phase was spherical. Then, the extruded samples were hot reshaped into 100 mm (length) × 100 mm (width) × 0.32 mm (thickness) precursor films using a press at 195 °C, before being isothermally crystallized at 130 °C for 20 min. During the hot-pressing process, the PP matrix melted and was reshaped, while the PA6 dispersed phase maintained its fibrillar or spherical shape. The films were then stretched longitudinally to specified strains of 20%, 50%, 100%, and 200% at 100 °C in a high-temperature chamber of the SANS CMT4104 tensile testing machine. Biaxial stretching consisted of longitudinal stretching to 200% at 100 °C, followed by transverse stretching to 200% at 125 °C. All stretching operations were performed at a speed of 10 mm/min. Before stretching, the samples were heated for 8 min at stretching temperature.

### 2.3. Characterization and Testing

#### 2.3.1. X-ray Diffraction (XRD)

An X-ray diffractometer (Ultima IV, reflection mode) was used to obtain the XRD spectra to characterize the crystalline structure of the samples. The fraction of β crystals (*K_β, XRD_*) can be calculated according to the following equation [34]:(1)$$K_{\beta ,XRD}=\frac{A_{\beta \left(300\right)}}{A_{\beta \left(300\right)}+A_{\alpha \left(040\right)}+A_{\alpha \left(130\right)}+A_{\alpha \left(110\right)}}\times 100\%$$
where *A_β_*_(300)_ was the area of the *β*(300) diffraction peak at 2θ = 16.1°. *A_α_*_(110)_, *A_α_*_(040)_, and *A_α_*_(130)_ were the areas of the fitted *α*(110), *α*(040), and *α*(130) diffraction peaks at 2θ = 14.1°, 16.9°, and 18.6°, respectively. The crystalline and amorphous peaks were fitted by JADE 6.0 software, and the corresponding peak areas were calculated.

#### 2.3.2. Scanning Electron Microscope (SEM)

The microstructural characteristics of the samples were observed at 10 kV using SEM (JEOL JSM-5900LV). β-PP was chemically etched at 60 °C for 24 h by a mixed acid solution with an etchant containing 1.3 wt.% potassium permanganate, 32.9 wt.% concentrated sulfuric acid, and 65.8 wt.% concentrated phosphoric acid. PA6 morphologies in PA6/PP(S) and PA6/PP(F) precursor films were observed after dissolving the PP by decalin at 160 °C for 6 h. Before observation, the samples were sprayed with gold to increase electrical conductivity.

#### 2.3.3. Two-Dimensional Wide-Angle X-ray Diffraction (2D-WAXD)

Two-dimensional-WAXD was performed on the HomeLab 2D-WAXD system (Rigaku, Tokyo, Japan). Before the test, the beam center and detector-to-sample distance were calibrated using a silver behenate standard. The detector-to-sample distance was 68 mm. Fit2D (open source) was used to extract 1D-WAXD curves from 2D-WAXD patterns. The structure of stretched samples acquired at a certain strain were frozen using liquid nitrogen. The crystal orientation was evaluated using Herman’s orientation factor (*f_H_*). For a particular crystal plane, Herman’s orientation function is defined as follows [35]:(2)$$f_{H}=\frac{3\langle cos^{2}\varphi \rangle_{hkl}-1}{2}$$
where *cos*^2^*φ* is the orientation factor defined as follows:(3)$$\langle cos^{2}\varphi \rangle_{hkl}=\frac{\int_{0}^{\pi /2}I\left(\varphi \right)cos^{2}\varphi sin\varphi d\varphi }{\int_{0}^{\pi /2}I\left(\varphi \right)sin\varphi d\varphi }$$
where *φ* is the azimuthal angle, and *I(φ)* is the scattered intensity along the angle *φ*.

#### 2.3.4. Atomic Force Microscopy (AFM)

The Anasys Afm+ system (Snata Barbara, CA 93101, USA) was used to analyze the surface of the microporous membranes in tapping mode, with an oscillation frequency of 100 kHz and a tip radius < 10 nm to allow imaging of small-scale features.

#### 2.3.5. Membrane Porosity and Pore Size Distribution

A capillary flow porometer (3H-2000PB, BeShiDe Inc., Beijing, China) was utilized to test the pore size distribution of the microporous membranes [36,37,38,39] under ASTM F316. The porosity (A_k_) of the membranes was determined using the n-butanol adsorption method. It was calculated by dividing the total volume of the membranes by the volume of the micropores. The test was performed by immersing microporous membranes in n-butanol for 6 h. Filter paper was then used to remove the n-butanol on the membrane surface carefully, and a precise balance was used to obtain the exact weight gain of the membranes after adsorption. The porosity of the microporous membrane was calculated from the following formula [40,41]:(4)$$A_{k}=\frac{w_{\mathrm{butanol}}/\rho_{\mathrm{butanol}}}{w_{0}/\rho_{0}+w_{\mathrm{butanol}}/\rho_{\mathrm{butanol}}}$$
where w_0_ and ρ_0_ were the weight and density of β-PP and PA6/PP blends, which were measured by an automatic densimeter (MDY350, Zhengzhou Huazhi Electronic Technology Co., LTD, Zhengzhou, China). w_butanol_ was the weight gain after immersion in n-butanol, and ρ_butanol_ was 0.808 g/cm^3^.

#### 2.3.6. Tensile Strength and Puncture Strength

The tensile and puncture strength were determined at 25 °C using a universal material testing machine (SANS CMT4104). Standard dumbbell bars were made for the tensile strength test performed according to the ISO 527-3 standard, with a crosshead speed of 20 mm/min. The gauge length was 25 cm, and no extensometer was used in the tensile test. Puncture tests were performed with a stainless-steel ball and fixture diameter of 2 mm and 30 mm, respectively, at a normal force loading speed of 50 mm/min. At least five parallel samples were used in each test.

#### 2.3.7. Light Microscopy Observation (OM)

A light microscope (BX51, Olympus, Tokyo, Japan) was used to observe the morphologies of the microporous membranes at the puncture point in transmitted light mode after the puncture test.

## 3. Results and Discussion

### 3.1. Structure Characterization of Precursor Films

The inner microstructure of the precursor films is critical to the microporous formation of stretched PP membranes [32]. Therefore, the crystal morphology of β-PP and the PA6 phase structure was examined by SEM first, as shown in Figure 1a–c. A high content of β crystals in the bundle shape was observed in β-PP (Figure 1a), whereas the spherical PA6 phase and fibrillar PA6 phase were dispersed uniformly in the PP matrix of PA6/PP(S) and PA6/PP(F), respectively, as shown in Figure 1b,c.

A large number of β crystals was formed in β-PP because TMB-5 is an effective β-nucleation agent, and the optimal temperature for the growth of β-crystals is 130 °C [24,34]. For two-phase blends, the phase morphology depends not only on the two-phase ratio, compatibilizers, etc., but is also associated with the external force field [33,42,43,44]. Therefore, the PA6/PP blends with different PA6 morphologies, i.e., spherical PA6 and fibrillar PA6 structures, were well constructed by controlling the stretching force field. For PA6/PP(F), the PA6 phase was stretched into well-oriented fibers at a high draw ratio (about 100). The mechanism of in-situ fibrillation of the PA6 phase was that the local extension force produced by hot stretching was bigger than the resistance to the formation of PA6 fibers [33,45]. The SEM results also indicated that the secondary processing at 195 °C did not change the morphologies of the PA6 phase.

To obtain the β and α crystals’ contents, Figure 1d shows the XRD curves of three unstretched films. Both β-PP and PA6/PP(S) showed characteristic diffraction peaks at 2θ = 16.1° for the β(300) crystal plane and at 14.1°, 16.9°, and 18.6° for the α(110), α(040), and α(130) crystal planes, respectively. According to Equation (1), the β crystal content of β-PP and PA6/PP(S) was 81.2% and 17.4%, respectively. Consistent with this, the curve of PA6/PP(S) had a much smaller β(300) diffraction peak than β-PP.

Both the α and β phases existed in the β-PP and PA6/PP(S) samples. β-PP contained a large number of β crystals, much higher than PA6/PP(S). The XRD curve of PA6/PP(F) only showed the diffraction peaks of α crystals, indicating that PA6/PP(F) was only composed of α crystals. In addition, the characteristic diffraction peak of α(200) belonging to PA6 appeared in the XRD curves of PA6/PP(S) and PA6/PP(F) [46].

To observe the PA6 morphology more clearly, a hot decalin bath was used to dissolve the PP matrix and the PA6 phase was obtained (Figure 2a,b). The spherical PA6 particles in PA6/PP(S) were 771.1 nm in average diameter (Figure 2(a1)), while the PA6 fibers in PA6/PP(F) were 253.9 nm in average diameter, and up to several tens of microns in length (Figure 2(b1)). These results indicated that the diameter distribution of both PA6 spherical particles and fibers was very uniform.

### 3.2. Micropore-Forming Process during Longitudinal Stretching

β-PP, PA6/PP(S), and PA6/PP(F) were uniaxially stretched to 20%, 50%, 100%, and 200% at 100 °C, respectively. The SEM images of β-PP at the early stage of deformation (ε = 20%) are given in Figure 3. Figure 3a exhibits some microcracks perpendicular to the stretching direction. Subsequently, to clarify the relationship between these deformation bands and β crystals, the amorphous regions of β-PP were etched away, as shown in Figure 3b. As marked by the yellow ellipses, these deformation bands occurred at the interfaces of the β crystals, indicating that the generation of initial microcracks was closely related to the separation of β crystals. Previous research has demonstrated that the amorphous area of β-PP has fewer tie chains than that of α-PP. The lack of a structure for stress transfer between β-lamellae results in the formation of a weak interface; thus, β-lamellae are easily separated under applied stress, and microporous structures in β-PP are formed with interface debonding [18].

The β-PP films were stretched longitudinally at 100 °C to the different tensile strains of 20%, 50%, 100%, and 200%, and the surface and cross-sectional microporous structures of the stretched membranes were subsequently characterized by SEM, as shown in Figure 4. First, the initial microcracks were formed at ε = 20% due to the separation of β-crystal defects (Figure 4(a1)), which further expanded at ε = 50% (Figure 4(a2)). At first, the microcracks were perpendicular to the tensile direction. As the strain increased to 100% and 200%, the direction of the micropores shifted to along the tensile direction (Figure 4(a3,a4)). Some long cracks along the load direction formed, accompanied by the fragmentation of the PP matrix. However, there were still many nonporous areas at a high strain of 200% (Figure 4(a4)), due to the formation of fewer initial microcracks (Figure 4(a1)). The internal microporous structure of β-PP was reflected by the cross-sectional SEM images, as shown in Figure 4(b1–b4). It can be observed that the structural characteristics of the internal micropores were consistent with those of the surface micropores. First, at ε = 20% and 50%, microcracks perpendicular to the stretching direction appeared, as shown in Figure 4(b1,b2). Then, the orientation direction of the micropores was shifted to along the stretching direction (Figure 4(b3,b4)). A relatively small number of locally distributed micropores can be observed at ε = 200%. The reason for this was that the defects within the β crystals resulted in fewer initial microcracks, which was insufficient to generate a large number of micropores.

PA6/PP(S) films were stretched longitudinally at 100 °C to different tensile strains of 20%, 50%, 100%, and 200%, and the surface and cross-sectional microporous structures of the stretched membranes were subsequently characterized by SEM, as shown in Figure 5. As shown in Figure 5(a1,a2), more microcracks appeared in PA6/PP(S) compared with β-PP (ε = 20% and 50%). The direction of surface microcracks gradually shifted to the stretching direction at ε = 100% and 200%, as shown in Figure 5(a3,a4). The number and size of micropores greatly increased. Furthermore, the inner microporous structure at different strains is presented in Figure 5(b1–b4) by cross-sectional SEM micrographs. It can be observed that the PA6 phase was spherical and did not deform as the strain increased. At ε = 20% and 50%, as shown in Figure 5(b1,b2), gaps between the PA6 particles and the PP matrix appeared, indicating the occurrence of phase interface separation. Micropores appeared in the surrounding PP matrix with increasing strain, and gradually elongated by the stretching force. The orientation of the micropores significantly improved at higher strains (ε = 100% and 200%), as shown in Figure 5(b3,b4). No severe interface debonding occurred between PA6 and PP.

PA6/PP(F) films were stretched longitudinally at 100 °C to different tensile strains of 20%, 50%, 100%, and 200%, and the surface and cross-sectional microporous structure of the stretched membranes were subsequently characterized by SEM, as shown in Figure 6. Before stretching, PA6 fibers in PA6/PP(F) ran along the direction of stretching. First, a large number of tiny microcracks appeared at ε = 20%, as shown in Figure 6(a1). The size and number of the microcracks gradually increased as the strain increased, as shown in Figure 6(a2–a4). At high strains of 100% and 200%, as shown in Figure 6(a3,a4), it can be observed that the PA6 fibers remained in the initial state and oriented along the stretching direction, while the PP matrix among these PA6 fibers was stretched into numerous micropores. Moreover, the cross-sectional microporous morphologies of PA6/PP(F) under different strains are presented by SEM images in Figure 6(b1–b4). At ε = 20% and 50%, only a few micropores appeared and the PA6 fibers maintained a good bond with the PP matrix, as shown in Figure 6(b1,b2). At ε = 100%, more micropores appeared, as shown in Figure 6(b3), since more separation of the interface between the PA6 fibers and PP occurred. As the strain increased to 200%, a highly oriented microporous structure appeared in the PP matrix among these parallel PA6 fibers, as shown in Figure 6(b4).

In summary, β-PP, PA6/PP(S), and PA6/PP(F) exhibited different mechanisms of micropore formation in the process of longitudinal stretching. Firstly, it has been proven that the mechanism of micropore formation of β-PP was the β-crystal defect debonding mechanism. Secondly, the β-crystal content of PA6/PP(S) and PA6/PP(F) unstretched films was 17.4% and 0%. However, more micropores formed in PA6/PP(S) and PA6/PP(F) than in β-PP, which suggested that the initial microcracks in PA6/PP(S) and PA6/PP(F) could only be derived from the phase interface separation of PA6 and PP, and could not have originated from the very few β crystals. Thirdly, the SEM results confirmed that the micropore-forming mechanism of PA6/PP(S) and PA6/PP(F) was two-phase interface separation. The small-sized PA6 phase formed a very large number of phase interfaces in the PP matrix. Under the tensile force, a significant stress concentration was generated at the interface between PA6 and PP, resulting in the formation of numerous micropores due to interfacial debonding, accompanied by the fragmentation and fibrillation of the PP matrix. Therefore, the micropore-forming mechanism in PA6/PP(S) and PA6/PP(F) was that initial micropores originated from phase interface separation. More initial microcracks were created by the phase interface separation, resulting in more micropores under greater deformation.

To clarify the phase transition and lamellar orientation, Figure 7 shows the 2D-WAXD results obtained from β-PP, PA6/PP(S), and PA6/PP(F) films at the four given strains of 0%, 50%, 100%, and 200% at 100 °C. Before stretching, the diffraction of β(300) in β-PP was very strong. Subsequently, the diffraction intensity from β(300) gradually decreased as a result of the β-α crystal transition under deformation. In contrast, the intensity of β(300) in PA6/PP(S) was weaker (ε = 0%) and gradually decreased to almost nothing at ε = 200%. The diffraction from β(300) in PA6/PP(F) cannot be observed due to the absence of β crystals. Moreover, it can be observed that the α(110), α(040), and α(130) diffractions of all the samples focused gradually on the horizontal direction with the strain increase, indicating that α crystals were gradually oriented. At ε = 200%, the diffraction arc intensity of PA6/PP(S) in the equatorial direction was the largest among the three samples, indicating that the largest crystal orientation was obtained for PA6/PP(S).

Figure 8 shows the 1D-WAXD curves for the three samples at different strains, which were integrated by the 2D-WAXD patterns. The β-PP and PA6/PP(S) unstretched samples exhibited α(110), β(300), α(040), α(130), β(311)/α(131), and α(041) diffraction peaks located at 2θ = 14.1°, 16.1°, 16.9°, 18.6°, 21.1°, and 21.8°, respectively. The strong diffraction peak of the β(300) plane of β-PP decreased rapidly under stretching force due to the β-α crystal transition, accompanied by β-crystal fragmentation (Figure 8a). The initial diffraction peak of β(300) in PA6/PP(S) was much smaller than β-PP (Figure 8b), while the diffraction peak of β(300) was invisible in PA6/PP(F) (Figure 8c). The small diffraction peaks of α(200) from PA6 can be observed in PA6/PP(S) and PA6/PP(F). Due to the fragmentation of the crystals, all the diffraction peaks decreased as the strain increased. The azimuthal intensity distribution curves and corresponding Herman’s orientation factor of α(110) diffraction at ε = 200% are presented in Figure 8d. PA6/PP(S) had a sharper peak than β-PP and PA6/PP(F). In terms of Herman’s orientation factor, PA6/PP(S) achieved the biggest crystal orientation, while the crystal orientation PA6/PP(F) at ε = 200% was the smallest.

### 3.3. Micropore-Forming Process during Transverse Stretching

Figure 9 shows SEM images of the surface microporous structure of β-PP, PA6/PP (S), and PA6/PP (F) during transverse stretching. At the strain of 200% × 0%, β-PP had more nonporous areas and fewer micropores than PA6/PP(S), as shown in Figure 9(a1,b1), while PA6/PP(F) contained a large number of parallel-aligned PA6 fibers with some tiny micropores among them, as shown in Figure 9(c1). In β-PP, the PP matrix that did not become porous during longitudinal stretching was also unable to become porous during transverse stretching. As shown in Figure 9(a2,a3), quite a lot of blocky nonporous areas appeared at strains of 200% × 50% and 200% × 100%. Then, at a maximum strain of 200% × 200%, a large number of coarse fibers were eventually generated, as shown in Figure 9(a4). Fewer nonporous areas and more micropores formed during longitudinal stretching in PA6/PP(S) than β-PP. Therefore, more micropores formed in PA6/PP(S) during transverse stretching than β-PP, and some PA6 spherical particles gradually appeared at the strain of 200% × 50% (Figure 9(b2)). As shown in Figure 9(b3), more and more PA6 spherical particles appeared and acted as anchor points, causing the surrounding PP matrix to transform into an excellent PP fiber network, with PA6 as the center, at the strain of 200% × 100%. At the maximum strain of 200% × 200%, an excellent spider-web-like microporous structure formed (Figure 9(b4)), accompanied by a dramatic increase in the micropores’ size and number. For PA6/PP(F), the parallel-aligned PA6 fibers were subjected to a vertical tensile force and gradually separated at the strain of 200% × 50%, as shown in Figure 9(c2). At the strain of 200% × 100%, further separation of PA6 fibers promoted fibrillation and micropore formation of the PP matrix, as shown in Figure 9(c3). At a maximum strain of 200% × 200%, the violent separation of PA6 fibers resulted in numerous tiny micropores and a dense fiber network composed of PA6 fibers interwoven with PP fibers, as shown in Figure 9(c4).

To summarize, the three membranes exhibited different pore-forming behaviors and final microporous structures during transverse stretching, due to the addition of two distinct morphologies of the PA6 phase. The reasons can be classified into two categories. Firstly, the microporous structure of the three membranes after longitudinal stretching was quite different. Secondly, the phase interface between PA6 and PP was critical in promoting the fragmentation of the PP matrix and the formation of fine fibers and additional micropores. As a result, PA6/PP(S) and PA6/PP(F) had superior micropore-forming ability to β-PP, resulting in an optimized microporous structure.

Figure 10 showed the variations in porosity of β-PP, PA6/PP(S), and PA6/PP(F) during sequential biaxial stretching. For β-PP, the porosity gradually increased with the strain and finally reached a maximum value of 52.7%. PA6/PP(S) showed the largest increase in porosity with strain increase because the interface between PA6 spherical particles and PP gradually pulled apart. Finally, the porosity reached the maximum value of 72.8%. For PA6/PP(F), since the PA6 fibers in PA6/PP(F) were more closely arranged during longitudinal stretching, only a relatively small number of micropores formed. The porosity of PA6/PP(F) was only 22.9% when longitudinally stretched to 200%, which was the lowest among the three membranes. At the strain of 200% × 50%, the porosity of PA6/PP(F) increased to 52.7% because the interface between PA6 fiber and PP pulled apart. The porosity of PA6/PP(F) eventually reached the maximum value of 63.1% at the strain of 200% × 200%. These results were consistent with the SEM results in Figure 9. The PA6/PP(S) and PA6/PP(F) samples presented superior pore-forming properties and increased porosities compared to β-PP, due to the micropore-forming mechanism of two-phase interface separation.

### 3.4. Structure and Performance Analysis of Biaxially Stretched Microporous Membranes (200% × 200%)

The detailed surface and cross-section microporous structure of β-PP, PA6/PP(S), and PA6/PP(F) membranes (200% × 200%) are shown in Figure 11. As shown in Figure 11a, the β-PP membrane revealed a typical dendritic microporous structure containing some coarse PP fibers. The PA6/PP(S) membrane displayed an excellent spider-web-like microporous structure centered on spherical PA6 particles, as observed in Figure 11b. Figure 11c shows that the PA6/PP(F) membrane exhibited a dendritic microporous structure, consisting of many thin PP fibers interwoven with PA6 fibers and smaller micropores. Figure 11(a1,b1,c1) shows the cross-sectional SEM micrographs of three membranes. The β-PP membrane showed a relatively dense multilayered structure (Figure 11(a1)). The PA6/PP(S) membrane showed that some PA6 spheric al particles were embedded into the microporous network (Figure 11(b1)). The PA6/PP(F) membranes showed that some PA6 fibers were embedded in the layered structure (Figure 11(c1)). PA6/PP(S) and PA6/PP(F) exhibited looser microporous structures than β-PP, which originated from their unique pore-forming mechanisms. It can be concluded that the interfacial separation between PA6 and PP promoted the fragmentation and fibrillation of the PP matrix, reduced the number of coarse PP fibers, and created a more optimized microporous structure than β-PP.

Figure 12 shows AFM height images and three-dimensional AFM images of the three membranes. It can be observed that β-PP presented a microporous network consisting of many coarse PP fibers, as shown in Figure 12a. PA6/PP(S) showed a spider-web-like microporous structure centered on some PA6 spherical particles and a larger pore size and looser PP fiber network than β-PP, as shown in Figure 12b. Since the PA6 fibers interweaved with PP fibers, the PA6/PP(F) membrane exhibited a thicker but uniform fiber network compared to the β-PP and PA6/PP(S), as displayed in Figure 12c. The surface 3D images and roughness are presented in Figure 12d–f. All the membranes exhibited relatively flat surfaces. PA6/PP(S) showed a slightly rougher surface than β-PP and PA6/PP(F), due to some raised PA6 particles.

To quantify their surface microporous structure, SEM micrographs of the three membranes were converted into black-and-white bicolor images using Image J (open sources), as shown in Figure 13a–c. The black area represents the polymer matrix, and the white area represents the micropores. It was calculated that the surface porosity increased from 12.24% for β-PP to 36.35% for PA6/PP(S) and 44.17% for PA6/PP(F). PA6/PP(S) and PA6/PP(F) exhibited a narrower surface pore area distribution than β-PP, as shown in Figure 13d. These results also reflected that PA6/PP(S) and PA6/PP(F) membranes exhibited a better microporous structure than β-PP.

Figure 14 shows the pore size distribution of the three membranes characterized by capillary flow porometry under ASTM F316. The pore size distribution of the PA6/PP(S) and PA6/PP(F) samples were both narrower than the β-PP samples. As summarized in Table 2, β-PP, PA6/PP(S), and PA6/PP(F) had 0.0643 μm, 0.0685 μm, and 0.0603 μm average pore sizes, respectively. The porosities of PA6/PP(S) and PA6/PP(F) were 72.8% and 63.1%, while β-PP had a smaller porosity of 52.7%. The PA6/PP(S) and PA6/PP(F) membranes showed 172.61 ± 23.57 L/(m^2^·h·bar) and 146.02 ± 20.12 L/(m^2^·h·bar) in pure water flux, while the β-PP membrane exhibited the smallest pure water flux of 69.29 ± 14.33 L/(m^2^·h·bar), due to the lower porosity and inferior micropore structure.

The microporous membranes’ high mechanical strength is a critical requirement for commercialization. As shown in Table 2, the tensile strengths of PA6/PP(S) and PA6/PP(F) in the MD direction were 74.3 ± 11.3 MPa and 57.3 ± 3.4 MPa, respectively, which were 13.0% and 32.9% lower compared to 85.4 ± 2.0 MPa of β-PP. The tensile strengths of PA6/PP(S) and PA6/PP(F) in the TD direction were 53.0 ± 2.1 MPa and 52.7 ± 1.4 MPa, respectively, which were slightly higher compared to 49.5 ± 0.5 MPa for β-PP. The puncture strengths of β-PP, PA6/PP(S), and PA6/PP(F) are given in Figure 15a and Table 2. The puncture strength of PA6/PP(F) was 94.8 ± 4.9 N/mm, which was slightly higher than that of β-PP (90.4 ± 3.7 N/mm), while the puncture strength of PA6/PP(S) was 65.75 ± 6.9 N/mm, which was 27.3% lower than that of β-PP.

The lower tensile strengths of PA6/PP(S) and PA6/PP(F) were presumed to be due to the presence of a weak interface between PA6 and PP. The relatively lower puncture strength of PA6/PP(S) was probably due to the weak PA6–PP interface defect and high porosity of 72.8%. The optical microscope photographs of puncture damage in Figure 15b–d demonstrated that the three membranes had different morphologies of puncture damage. PA6/PP(S) showed a long tear-like fracture, indicating that the membrane was easily deformed and torn by external forces. β-PP and PA6/PP(F) showed a hole and a short crack, respectively, indicating that the membranes showed a higher resistance to puncture damage.

### 3.5. Micropore-Forming Mechanism of Biaxially Stretched Microporous Membranes

Figure 16 shows the probable micropore-forming mechanisms of the three membranes during sequential biaxial stretching. Firstly, the micropore-forming mechanism of β-PP was the debonding of β-lamellae. The β-crystal content of β-PP unstretched films was 81.2%. The β crystals were randomly arranged (Figure 16(a1)). The defects of the β-crystals can only be transformed into a small amount of microporosity. Fewer micropores formed during longitudinal stretching (Figure 16(a2)). A microporous structure containing many coarse fibers and low porosity formed in transverse stretching (Figure 16(a3)). Secondly, the micropore-forming mechanism of PA6/PP(S) was the debonding of the PA6 spherical particles–PP interface. The PA6 spherical particles were well dispersed in the PP matrix (Figure 16(b1)). During longitudinal stretching, the interface between the PA6 spherical particles and the PP matrix was deformed, and an oriented microporous structure formed at the interface (Figure 16(b2)). Subsequently, the phase interface and micropores were further expanded upon transverse stretching, which promoted the fragmentation and fibrillation of the PP matrix and formed a spider-web-like microporous structure centered on PA6 spherical particles. Thus, it exhibited an optimized microporous structure and increased porosity of 72.8% after biaxial stretching (Figure 16(b3)). Thirdly, the micropore-forming mechanism of PA6/PP(F) was PA6 fibers–PP interface separation. Before stretching, precursor films contained many parallel-arranged PA6 fibers (Figure 16(c1)). Only a small amount of micropores formed after longitudinal stretching, since the stretching direction was parallel to the orientation direction of PA6 fibers (Figure 16(c2)). As shown in Figure 16(c3), the transverse stretching caused the violent separation of PA6 fibers, resulting in a fine microporous structure of PA6 fibers interwoven with PP fibers and high porosity of 63.1%. In summary, as a result of the unique microporous formation mechanism of the separation of the interface between PA6 and PP, the PA6/PP(S) and PA6/PP(F) biaxial stretched membranes achieved an optimized pore structure, high porosity, narrow pore size distribution, and high permeate flux.

## 4. Conclusions

PA6/PP(S) and PA6/PP(F) biaxial stretching membranes showed different micropore-forming mechanisms, as confirmed by SEM and the porosity test. Our conclusions are as follows:

(1) PA6/PP blends containing uniformly distributed spherical and fibrous PA6 phases were successfully prepared by controlling the stretch ratio in the extrusion process, thereby constructing different two-phase interface morphologies.

(2) During the longitudinal stretching, more microcracks originating from PA6-PP interfacial debonding were generated in PA6/PP(S) and PA6/PP(F) compared to β-PP. The two-phase interface and micropores in PA6/PP(S) further expanded during transverse stretching, forming a spider-web-like microporous structure centered on PA6 spherical particles. The parallel-aligned PA6 fibers in PA6/PP(F) separated violently during transverse stretching, resulting in the extensive fragmentation and fibrillation of PP, and causing an excellent microporous network of PA6 fibers interwoven with PP fibers.

(3) The prepared PA6/PP(S) and PA6/PP(F) biaxially stretched membranes presented a better microporous structure than β-PP, thus exhibiting a higher porosity, more uniform pore size distribution, and better permeability.

## Figures and Tables

**Figure 1 polymers-14-02291-f001:**
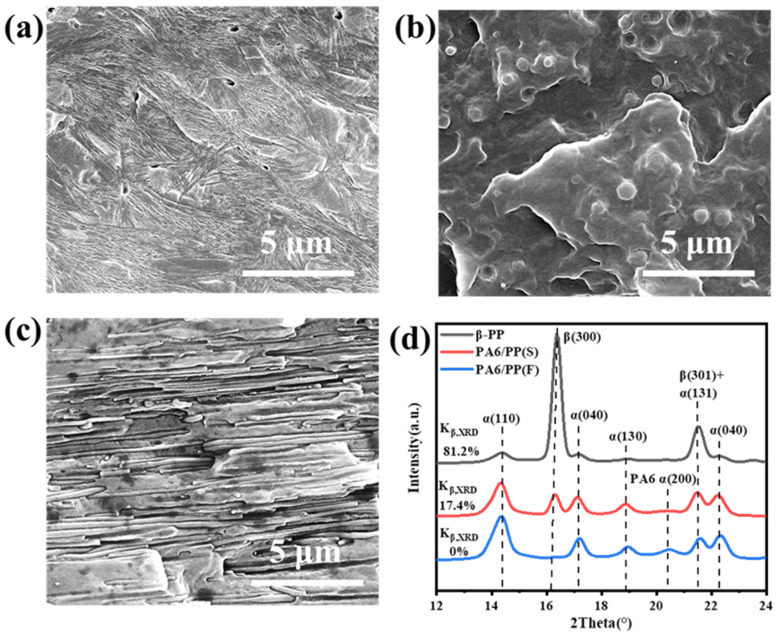
(**a**) Cross-sectional SEM images of β-PP unstretched films, with the amorphous area etched out. The cryo-fractured surface SEM images of (**b**) PA6/PP(S) and (**c**) PA6/PP(F) unstretched films. (**d**) XRD curves of three unstretched films.

**Figure 2 polymers-14-02291-f002:**
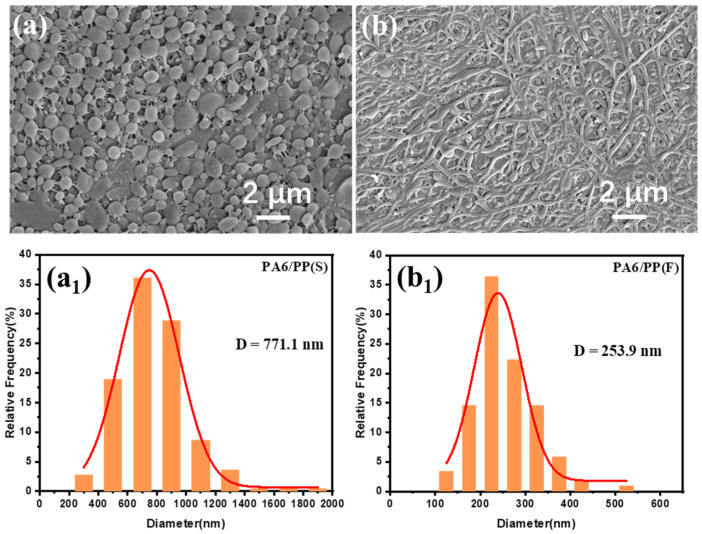
SEM images of (**a**) PA6/PP(S) and (**b**) PA6/PP(F) films after high-temperature treatment with decalin to remove the PP phase. The diameter distribution curves and average diameter (D) of PA6 spherical particles and fibers are shown in (**a1**,**b1**).

**Figure 3 polymers-14-02291-f003:**
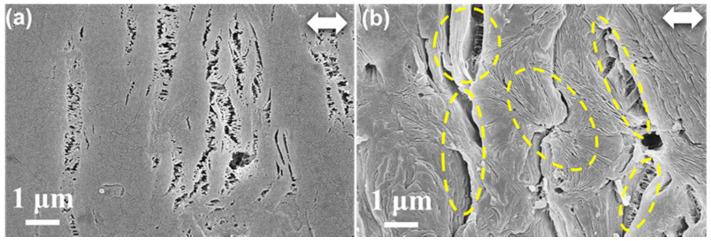
(**a**) SEM image of β-PP stretched to 20% at 100 °C. (**b**) SEM image of β-PP stretched to 20% at 100 °C, followed by etching away of the amorphous region. The white double arrow represents the direction of stretching.

**Figure 4 polymers-14-02291-f004:**
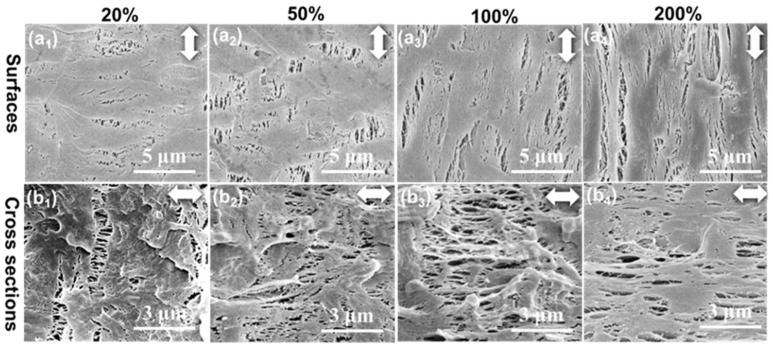
SEM images of the (**a1**–**a4**) surface and (**b1**–**b4**) cross-section microporous structure of β-PP films during longitudinal stretching. The tensile temperature was 100 °C and the tensile strains were (**a1**,**b1**) 20%, (**a2**,**b2**) 50%, (**a3**,**b3**) 100%, and (**a4**,**b4**) 200%. The white double arrow represents the direction of stretching.

**Figure 5 polymers-14-02291-f005:**
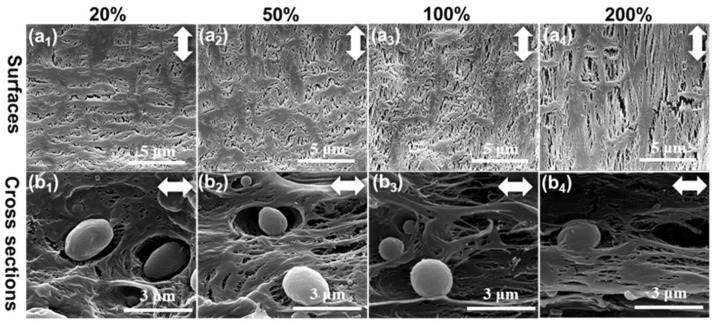
SEM images of the (**a1**–**a4**) surface and (**b1**–**b4**) cross-section microporous structure of PA6/PP(S) films during longitudinal stretching. The tensile temperature was 100 °C and the tensile strains were (**a1**,**b1**) 20%, (**a2**,**b2**) 50%, (**a3**,**b3**) 100%, and (**a4**,**b4**) 200%. The white double arrow represents the direction of stretching.

**Figure 6 polymers-14-02291-f006:**
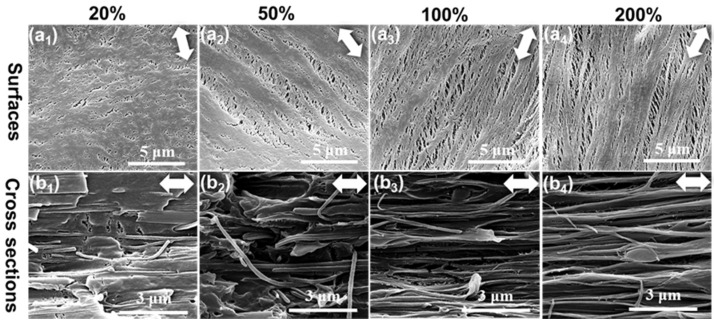
SEM images of the (**a1**–**a4**) surface and (**b1**–**b4**) cross-section microporous structure of PA6/PP(F) films during longitudinal stretching. The tensile temperature was 100 °C and the tensile strains were (**a1**,**b1**) 20%, (**a2**,**b2**) 50%, (**a3**,**b3**) 100%, and (**a4**,**b4**) 200%. The white double arrow represents the direction of stretching.

**Figure 7 polymers-14-02291-f007:**
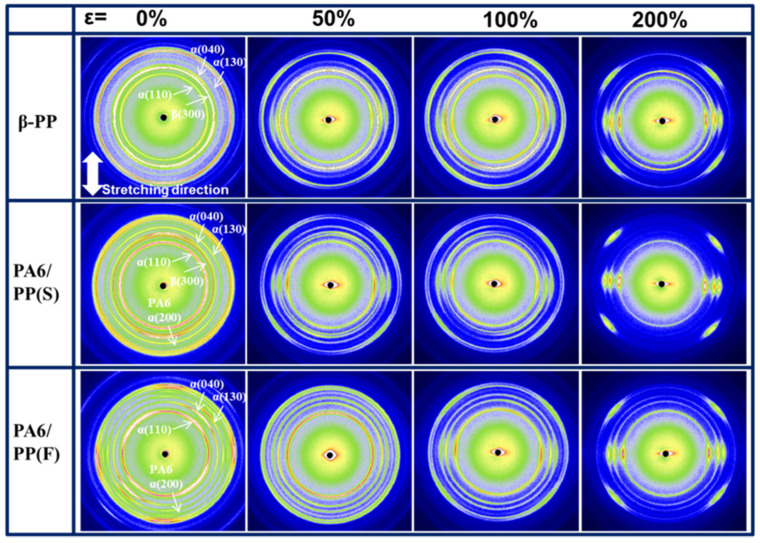
Two-dimensional-WAXD results of the structural evolution of longitudinally stretched β-PP, PA6/PP(S), and PA6/PP(F) films. The tensile temperature was 100 °C and the strains were 0%, 50%, 100%, and 200%.

**Figure 8 polymers-14-02291-f008:**
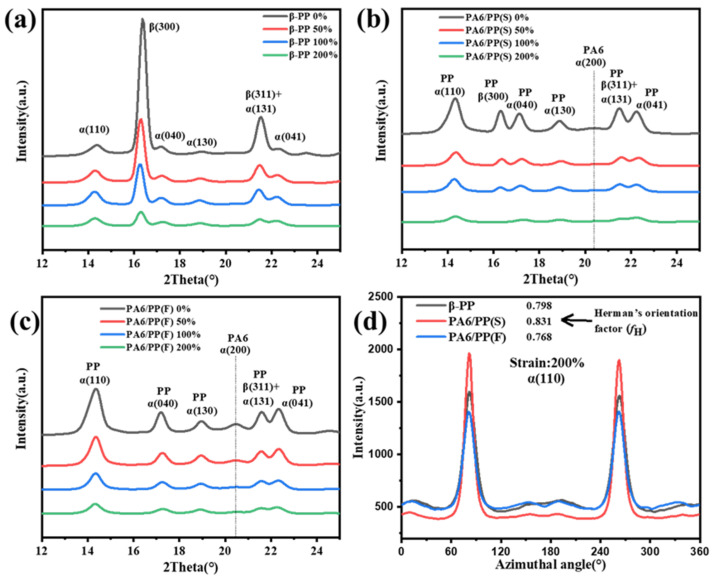
One-dimensional-WAXD curves integrated from two-dimensional-WAXD patterns of (**a**) β-PP, (**b**) PA6/PP(S), and (**c**) PA6/PP(F). (**d**) α(110) intensity distribution with the azimuthal angle at ε = 200%.

**Figure 9 polymers-14-02291-f009:**
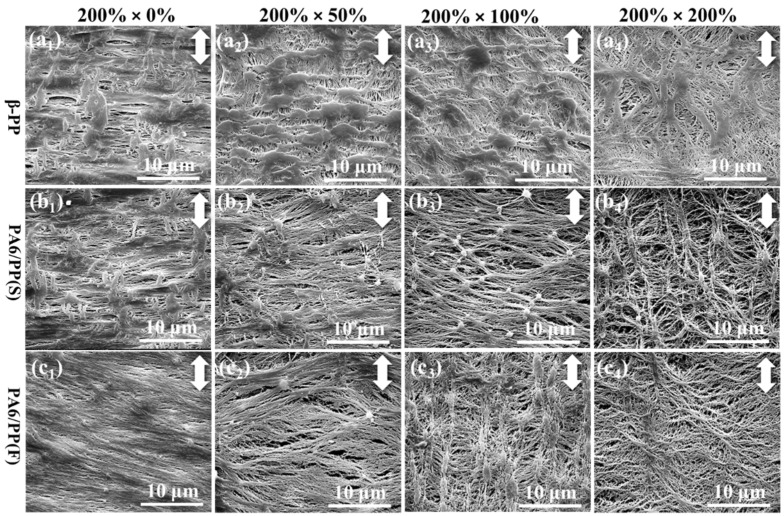
SEM images of the surface microporous structure of (**a**) β-PP, (**b**) PA6/PP(S) and (**c**) PA6/PP(F) during transverse stretching at 125 °C: (**a1**,**b1**,**c1**) 200% × 0%, (**a2**,**b2**,**c2**) 200% × 50%, (**a3**,**b3**,**c3**) 200% × 100%, (**a4**,**b4**,**c4**) 200% × 200%. The white double arrow represents the direction of stretching.

**Figure 10 polymers-14-02291-f010:**
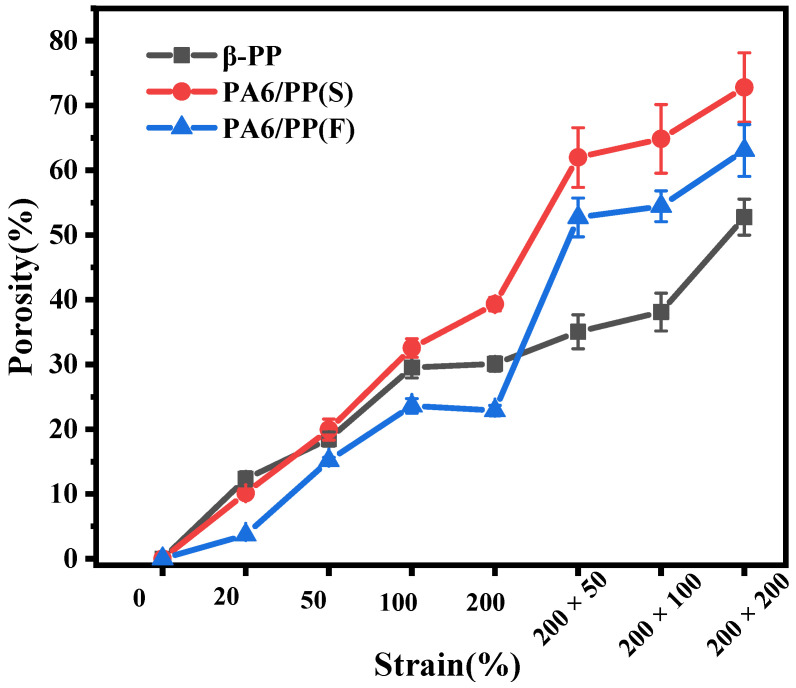
Variations in the porosity of β-PP, PA6/PP(S), and PA6/PP(F) membranes during the sequential biaxial stretching process.

**Figure 11 polymers-14-02291-f011:**
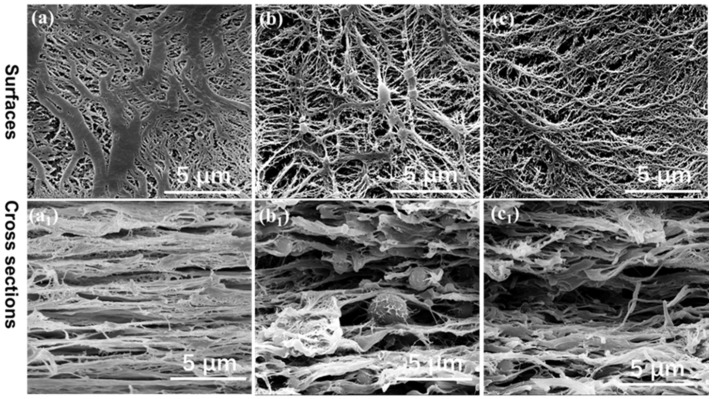
SEM images of the (**a**–**c**) surface and (**a1**,**b1**,**c1**) cross-section microporous structure of (**a**,**a1**) β-PP, (**b**,**b1**) PA6/PP(S), and (**c**,**c1**) PA6/PP(F). The strain was 200% × 200%.

**Figure 12 polymers-14-02291-f012:**
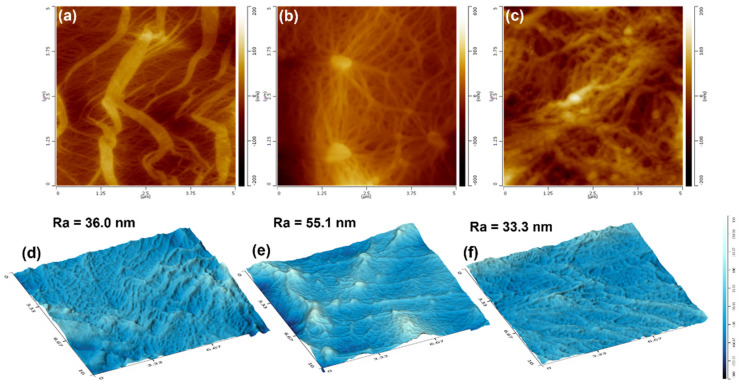
(**a**–**c**) AFM height images (5 μm × 5 μm) and (**d**–**f**) 3D AFM images (10 μm × 10 μm) of (**a**) β-PP, (**b**) PA6/PP(S), and (**c**) PA6/PP(F) microporous membranes. Ra represents the surface roughness of the microporous membranes.

**Figure 13 polymers-14-02291-f013:**
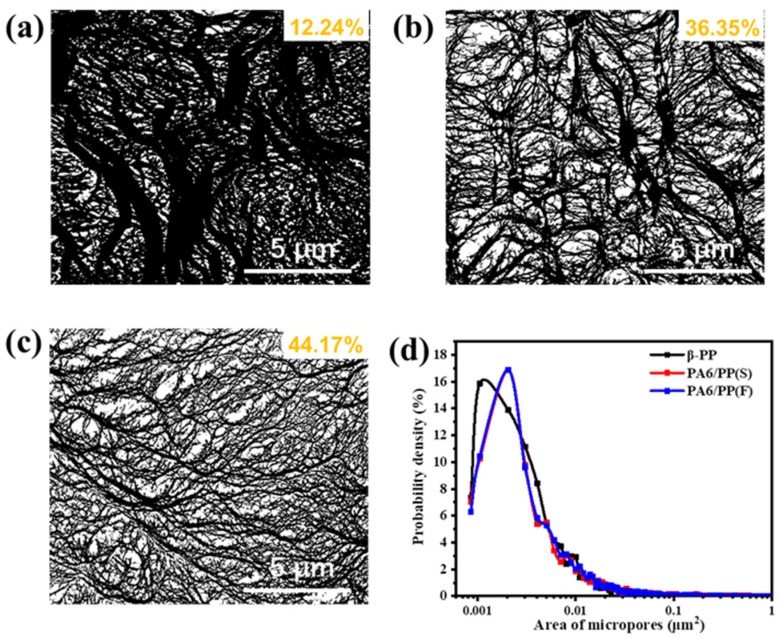
Microporous morphologies of (**a**) β-PP, (**b**) PA6/PP(S), and (**c**) PA6/PP(F) membranes processed using Image J software (the white part is the micropores and the black part is the PP matrix). The calculated surface porosity is indicated in the upper right corner of images. (**d**) The surface pore area distribution of the three microporous membranes.

**Figure 14 polymers-14-02291-f014:**
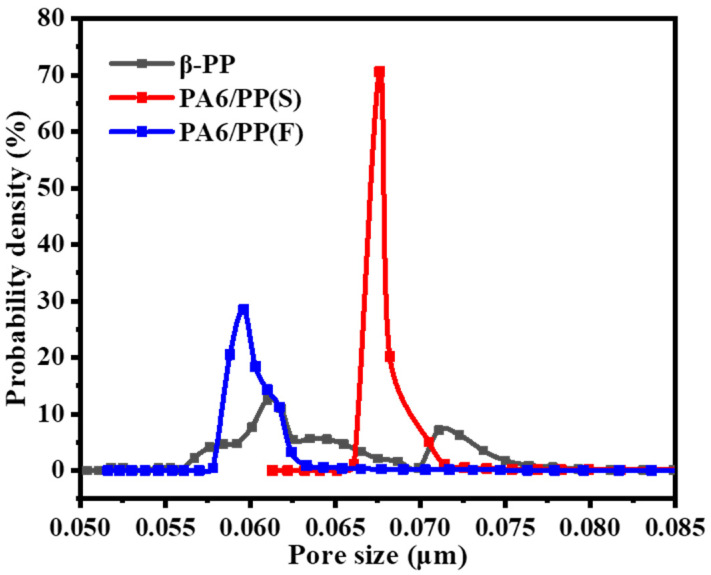
Pore size distribution curves of β-PP, PA6/PP(S), and PA6/PP(F) biaxial stretched membranes obtained via capillary flow porometry.

**Figure 15 polymers-14-02291-f015:**
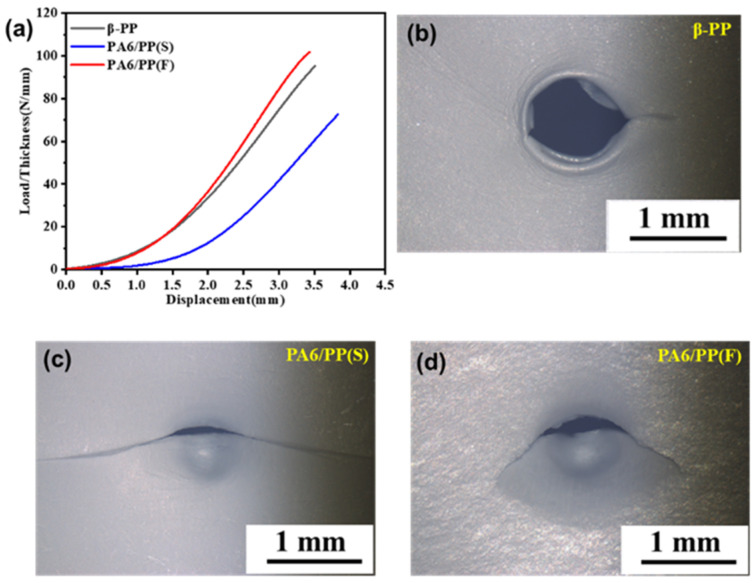
(**a**) The puncture property of β-PP, PA6/PP(S), and PA6/PP(F) microporous membranes; (**b**–**d**) the optical microscopic photographs of the puncture points after the puncture.

**Figure 16 polymers-14-02291-f016:**
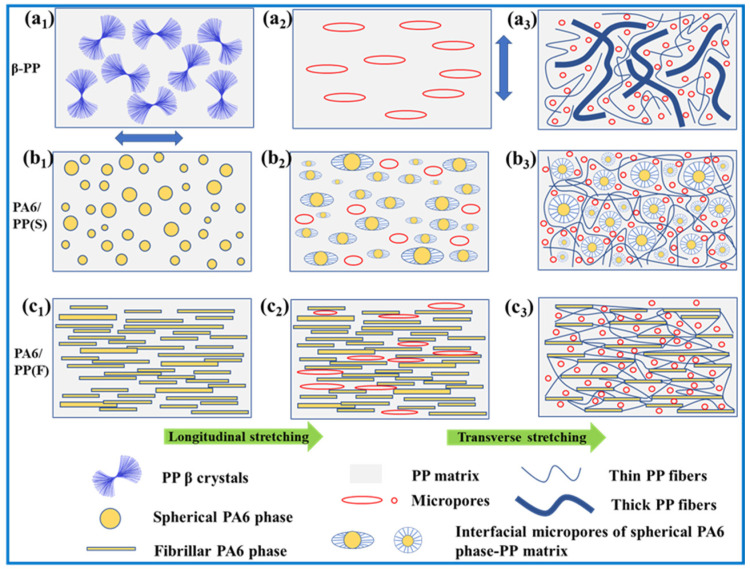
The schematic diagram of the micropore-forming mechanisms of (**a**) β-PP, (**b**) PA6/PP(S), and (**c**) PA6/PP(F) during sequential biaxial stretching. The blue double-headed arrow indicates the stretching direction. And (**a1**,**b1**,**c1**) indicate the structure of unstretched films; (**a2**,**b2**,**c2**) indicate the structure of longitudinally stretched membranes (200%); (**a3**,**b3**,**c3**) indicate the structure of biaxially stretched membranes (200% × 200%).

**Table 1 polymers-14-02291-t001:** Compositions of β-PP, PA6/PP(S), and PA6/PP(F) samples.

Title 1	iPP (wt.%)	PA6 (wt.%)	PPgMA (wt.%)	TMB-5 (wt.%)	Antioxidant 1010 (wt.%)
β-PP	99.2	0	0	0.5	0.3
PA6/PP(S)	76.2	18	5	0.5	0.3
PA6/PP(F)	76.2	18	5	0.5	0.3

**Table 2 polymers-14-02291-t002:** Thickness, average pore size, porosity, pure water flux, and mechanical properties of biaxially stretched membranes.

Samples	β-PP	PA6/PP(S)	PA6/PP(F)
Thickness (μm)	75 ± 4	80 ± 3	75 ± 5
Average pore size (μm)	0.0643	0.0685	0.0603
Porosity (%)	52.7 ± 2.8	72.8 ± 5.4	63.1 ± 4.0
Pure water flux (L/(m^2^·h·bar))	69.3 ± 14.3	172.6 ± 23.6	146.0 ± 20.1
Tensile strength-MD (MPa)	85.4 ± 2.0	74.3 ± 11.3	57.3 ± 3.4
Tensile strength-TD (MPa)	49.5 ± 0.5	53.0 ± 2.1	52.7 ± 1.4
Puncture strength (N/mm)	90.4 ± 3.7	65.8 ± 6.9	94.8 ± 4.9

## Data Availability

Data from this study are available upon request from the corresponding author.

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
