# Peer review of "Microporous Formation Mechanism of Biaxial Stretching PA6/PP Membranes with High Porosity and Uniform Pore Size Distribution"

_polymers, 2022, doi:10.3390/polym14112291_

Round 1

Reviewer 1 Report

The quality of this manuscript cannot be ranked based upon totally unacceptable presentation.It is totally inappropriate to present results and discussion together:please give results first and then discuss them with reference to the literature.Then the conclusion will be significant.

1) The main question addressed by the research is to investigate the microporous formation mechanism of biaxial stretching PA6/PP membranes with high porosity and uniform pore size distribution.

2) The topic s original and relevant in the field for PA6/PP albeit the microporous formation is mastered in case of ePTFE.

3) This publication takes advantage of many techniques  of analysis and a justification is given.However there is no statistical validation of the pore sizes and the volume of void.

4) The authors shall add clear cut statistical figures to compare with other microporous polymers.

5) The conclusion is too vague to e convincing.

6) I have a deep respect for the research conducted in China.However the bibliography as presented is deliberately provocative.

Once again,it is unacceptable to mix Results and Discussion to allow the readership to make its own opinion about the results and appreciate the discussion proposed by the authors.              

Reviewer 2 Report

Dear Authors,

I have performed the review of your manuscript. Some sections require to be discussed with detail (Materials and Methods, and Results and Discussion).

On the other hand, I found several issues about the English language, these issues were: word choise, wordy sentences, punctuation in compound/complex sentences, sentences to long (hard-to-read test), improper formating, etc, etc. These were highlighted in your manuscript.

I suggest that your manuscript must be revised by a native English speaker.

Round 2

Reviewer 2 Report

Dear Authors,

I have performed the review of your manuscript and I can see that you have modified the points highlighted in my previous report. I agree with this manuscript.

Sincerely yours,

The reviewer